# ^18^F-FDG-PET Imaging Patterns in Autoimmune Encephalitis: Impact of Image Analysis on the Results

**DOI:** 10.3390/diagnostics10060356

**Published:** 2020-05-29

**Authors:** David Moreno-Ajona, Elena Prieto, Fabiana Grisanti, Inés Esparragosa, Lizeth Sánchez Orduz, Jaime Gállego Pérez-Larraya, Javier Arbizu, Mario Riverol

**Affiliations:** 1Department of Neurology, Clínica Universidad de Navarra, Pío XII 36, 31008 Pamplona, Spain; dmoreno.1@unav.es (D.M.-A.); iesparragos@unav.es (I.E.); jgallego@unav.es (J.G.P.-L.); mriverol@unav.es (M.R.); 2Department of Nuclear Medicine, Clínica Universidad de Navarra, Pío XII 36, 31008 Pamplona, Spain; eprietoaz@unav.es (E.P.); fgrisanti@unav.es (F.G.); lizethsanchezorduz@gmail.com (L.S.O.); 3SPECT Medicina Nuclear S.A.S, UNAB, Bucaramanga 681004, Colombia

**Keywords:** ^18^F-FDG-PET, voxel-based analysis, assisted analysis, autoimmune encephalitis, limbic encephalitis

## Abstract

Brain positron emission tomography imaging with 18Fluorine-fluorodeoxyglucose (FDG-PET) has demonstrated utility in suspected autoimmune encephalitis. Visual and/or assisted image reading is not well established to evaluate hypometabolism/hypermetabolism. We retrospectively evaluated patients with autoimmune encephalitis between 2003 and 2018. Patients underwent EEG, brain magnetic resonance imaging (MRI), cerebrospinal fluid (CSF) sampling and autoantibodies testing. Individual FDG-PET images were evaluated by standard visual reading and assisted by voxel-based analyses, compared to a normal database. For the latter, three different methods were performed: two based on statistical surface projections (Siemens syngo.via Database Comparison, and 3D-SSP Neurostat) and one based on statistical parametric mapping (SPM12). Hypometabolic and hypermetabolic findings were grouped to identify specific patterns. We found six cases with definite diagnosis of autoimmune encephalitis. Two cases had anti-LGI1, one had anti-NMDA-R and two anti-CASPR2 antibodies, and one was seronegative. ^18^F-FDG-PET metabolic abnormalities were present in all cases, regardless of the method of analysis. Medial–temporal and extra-limbic hypermetabolism were more clearly depicted by voxel-based analyses. We found autoantibody-specific patterns in line with the literature. Statistical surface projection (SSP) methods (Neurostat and syngo.via Database Comparison) were more sensitive and localized larger hypermetabolic areas. As it may lead to comparable and accurate results, visual analysis of FDG-PET studies for the diagnosis of autoimmune encephalitis benefits from voxel-based analysis, beyond the approach based on MRI, CSF sample and EEG.

## 1. Introduction

Autoimmune encephalitis (AE) is an inflammatory disorder of the brain associated with neurologic dysfunction and is frequently a challenging diagnosis for the clinician. The pathogenesis of AE is related to the presence of autoantibodies against intracellular antigens (Hu, Ma2, GAD), synaptic receptors (NMDA receptor, AMPA receptor, GABA receptor, mGluR5, Dopamine receptor), ion channels and other cell-surface proteins (LGI-1, CASPR2, DPPX; MOG, AQP4, GQ1b). A specific type of AE is limbic encephalitis (LE), which was described for the first time in the 1960s [1]. In LE, inflammation affects predominantly the medial temporal lobes, and it may present with memory impairment, hallucinations, anxiety, irritability, depression, seizures and sleep alterations [2]. Treatment in patients with LE is often delayed due to the lack of specific symptoms and the time it takes to obtain the result of the autoantibodies analysis. A new clinical approach was proposed to treat subjects with a high clinical suspicion of AE, including LE, potentially leading to better outcomes [3]. This approach relies on neurological evaluation, brain magnetic resonance imaging (MRI) and cerebrospinal fluid (CSF) sampling to define possible AE. In these proposed guidelines, ^18^FDG-PET is mentioned as an alternative to MRI, however, only for the diagnosis of definite AE [3].

Several case reports that include positron emission tomography imaging with 18Fluorine-fluorodeoxyglucose (FDG-PET) were published in the late 1990s; the first was an anti-Hu AE in 1998 [4]. FDG-PET findings were discordant with MRI findings as they showed bilateral increased glucose metabolism in the temporal lobes, whereas MRI showed unilateral temporal lobe hyperintensity [4]. More recently, different case series showed poor concordance between both modalities [5,6,7]. Indeed, these studies reflected that FDG-PET imaging may be more sensitive than MRI in showing increased FDG metabolism in normal-appearing medial temporal lobes. In addition, good correlation between FDG-PET patterns and clinical presentation was observed [8]. Most importantly, when autoantibodies were negative and MRI findings were unremarkable, FDG-PET showed typical findings of AE [9].

Medial temporal lobe involvement has been traditionally associated with LE. This was consistently observed in the presence of classical paraneoplastic antibodies and anti-voltage gated potassium channel (VGKC) antibodies. Different patterns have been described depending on the patient’s age, with a neurodegenerative-like hypometabolism characteristically observed in the elderly [10,11]. Mesiotemporal hypermetabolism was seen in patients with positive intracellular antigens, probably due to a T-cell-mediated inflammatory process. On the other hand, in the presence of cell surface antibodies, hypometabolism was found. This is believed to be related to antibody-capping and subsequent receptor internalization [11]. Specifically, in anti-NMDA receptor encephalitis, an “anteroposterior gradient” was described, with frontal and temporal hypermetabolism associated with occipital hypometabolism [12,13,14,15,16,17,18,19]. Selective hypermetabolism of the basal ganglia was reported as a typical finding in anti-NMDA receptor encephalitis [17,18,19]. However, all these findings may depend on the method of FDG-PET analysis applied. These studies used two different methods of analyses. Namely, these were either standard visual reading or comparisons between groups of patients and healthy controls using voxel-based analysis. Regarding the latter, statistical parametric mapping (SPM) was the most commonly reported, with some methodological differences among studies [12,13,19].

The purpose of our study was to determine the additional value of voxel-based analyses methods to the standard visual reading of individual FDG-PET images.

## 2. Materials and Methods

### 2.1. Database Analysis

We performed a retrospective database analysis as part of an internal audit to identify patients evaluated between 2003 and 2018 with definite diagnosis of AE according to Graus and colleagues’ Position Paper [3]. We identified 6 patients in whom potential differential diagnoses were excluded by adequate tests, and who underwent autoantibodies analysis, MRI and electroencephalogram (EEG). Additionally, an early brain FDG-PET study was performed in all cases within the first week from the onset of symptoms. A whole-body –FDG-PET scan was added to the paraneoplastic workup of AE screening for malignancy in three patients (cases 1, 2 and 5). Patients fasted for at least 4 h before the study. Forty minutes after the injection of 336.7 ± 72.7 MBq of ^18^Fluorine-fluorodeoxyglucose, a 20-min PET/CT scan was acquired. All patients were scanned in a Siemens Biograph mCT TrueV. Acquisition and reconstruction were performed with the standard protocol for brain studies, as previously described [20]. Informed consent was obtained from all individual participants included in the study. Additional informed consent was obtained from all individual participants from whom identifiable information is included in this article.

### 2.2. FDG-PET Image Analysis

FDG-PET images of the brain were analyzed on an individual basis, using standard visual reading [21] and visual-assisted by voxel-based analysis. Areas of both hypermetabolism and hypometabolism in the FDG-PET scan of each patient were evaluated and agreed upon by three nuclear medicine physicians (JA, FG and LS) according to visual readings. This was followed by visual assisted analyses using the three methods: SPM12 (Wellcome Department of Cognitive Neurology, Institute of Neurology, London, UK) [22], statistical surface projections (SSP) with a normal database comparison by using the free access software Neurostat 3D-SSP [23] and the commercial software syngo.via Database Comparison provided by Siemens (Siemens Healthcare GmbH, Erlangen, Germany) [24]. These two methods include their own databases of normal subjects according to different age groups. As for SPM, a database of healthy controls from our site [25] was used to obtain individual-to-group differences. To this end, FDG-PET images were spatially normalized (using a specifically customized FDG template), intensity normalized to the pons region (predefined over Montreal Neurological Institute space) and spatially smoothed with a Gaussian kernel (12 mm). The pons is a region that has been widely used for activity normalization [23] and is considered to be unaffected by the pathology under study. As an exploratory approach, the threshold of the T-map images was set at two significance levels, *p* < 0.001 and *p* < 0.005 (uncorrected) with an extent threshold of 40 voxels. For the SSP methods areas above and below, two standard deviations (SD) were considered significant for hypermetabolism or hypometabolism.

All patients signed an informed consent form prior to submission, which was reviewed by the Research Ethics Committee of the University of Navarra Clinic.

## 3. Results

### 3.1. Clinical Findings

The study included six patients, three men and three women, with ages ranging from 17 to 78 years. Clinical features and complementary tests are summarized in Table 1.

Cognitive impairment was the first symptom in 5/6 cases with LE and behavioral changes were the first symptoms in the patient with anti-NMDA receptor encephalitis. Characteristic facio-brachial seizures occurred in the two anti-LGI-1 positive patients. The two anti-CASPR2 positive cases suffered from autonomic seizures consisting of recurrent second-lasting episodes of cold sensation and piloerection. The clinical course of the majority of the cases was subacute. However, a 17-year-old patient developed multiple recurrent acute episodes of behavioral disorders and hallucinations followed by status epilepticus, which led to the suspicion of an anti-NMDA receptor encephalitis.

Additionally, hyponatremia was found in serum analyses in both cases with anti-LGI-1 LE. When autoantibodies were found to be positive, these were positive in both CSF and blood. The brain MRI was considered abnormal in 4/6 individuals. Namely, medial temporal hyperintensity in the T2-FLAIR images was the most frequent finding and was associated with LGI-1 and CASPR2 antibodies.

On the other hand, anti-NMDA receptor and seronegative LE exhibited normal brain MRIs. The EEG showed focal temporal epileptiform discharges in 4/6 patients. The CSF initial analysis showed lymphocytic pleocytosis and mild hyperproteinorrachia was found in 3/6 patients.

Whole-body FDG-PET was performed in three patients to rule out a paraneoplastic etiology and was negative in all cases.

Immunotherapy was administered in all cases, including immunoglobulins 0.4 g/Kg/day for 5 days plus methylprednisolone (1 g/day for 5 days). Most patients required new treatment cycles as symptoms recurred. Two patients (one anti-LGI-1 and one anti-CASPR2) also required a rituximab cycle, which led to the resolution of symptoms.

### 3.2. Brain ^18^F-FDG-PET/CT Findings

Brain FDG-PET exhibited metabolic abnormalities in all cases, whereas MRI, CSF and EEG were all abnormal in 2/6 patients (Table 1). Standard visual analysis was limited when evaluating hypermetabolism. These findings were only evident when utilizing voxel-based analysis in both anti-CASPR2 cases and in one anti-LGI-1 (Table 2, case 2).

The global evaluation through voxel-based analyses showed hypermetabolism on the medial temporal lobe (MTL) as the main finding in all LE cases. However, SSP methods (Neurostat and syngo.via Database Comparison) were more sensitive and localized larger hypermetabolic areas than SPM in anti-LGI-1 cases (Table 2, cases 4 and 6). In cases 3 and 4, hypermetabolism was more evident in SPM when the threshold was adjusted to *p* < 0.005. Interestingly, in case 6 (anti-CASPR2), MTL hypermetabolism was not exhibited by SPM even when using *p* < 0.005 as the threshold. There were no differences between Neurostat and syngo.via Database Comparison.

Some extra-limbic abnormalities, which affected cortical and subcortical areas, were observed with different patterns depending on the autoantibodies involved. These were clearly depicted by the voxel-based analyses, whereas most of them were less evident with the standard visual reading.

Overall, SSP methods were superior in detecting both hypermetabolism as well as hypometabolism (see Table 2). SPM was limited to showing the characteristic basal ganglia hypermetabolism in case 4 (anti-LGI-1). Both anti-LGI-1 cases depicted the most sparing pattern, with hypermetabolism in basal ganglia and cerebellum, coexisting with hypometabolism in frontal and posterior association cortex including posterior cingulate hypometabolism (Figure 1).

In contrast, more widespread patterns involving both hypermetabolic and hypometabolic cortical areas were shown in the anti-NMAD receptor and the seronegative cases. The anti-NMDA receptor encephalitis (case 1) showed an antero-posterior gradient, with motor cortex hypometabolism as well as hyperactivity of the left temporal fusiform, bilateral parietal and posterior cingulate cortex (Figure 2).

Both cases with anti-CASPR2 LE showed comparable findings in MRI and FDG-PET images, including both standard and voxel-based analyses. Consistently, bilateral amygdalo-hippocampal hyperintensity correlated with hypermetabolic areas (Figure 3). Notably, none of the cases were on sedative, anesthetic or antiepileptic drugs at the time of the FDG-PET scan.

### 3.3. Final Diagnosis after Applying the Proposed Clinical Approach

The proposed diagnostic criteria for possible AE comprise a compatible clinical picture: new focal CNS findings/new seizures/CSF pleocytosis/suggestive MRI and exclusion of alternative causes [3]. For definite AE, it is required to find T2-weighted fluid-attenuated inversion recovery MRI restricted to MTL bilaterally or compatible FDG-PET, as well as a compatible EEG or CSF pleocytosis. In our series, MRI, CSF and EEG were all abnormal in 2/6 patients. Five out of six would fit the criteria for possible AE, whereas 6/6 would fit the criteria for definite AE only when using brain FDG-PET, as two cases showed no brain MRI abnormalities.

## 4. Discussion

FDG-PET abnormalities were more evident when utilizing voxel-based analyses as a complementary tool for standard visual reading. Voxel-based analyses detected MTL and extra limbic hypermetabolism, as well as hypometabolism, while the SSP methods were slightly more sensitive than SPM, but with no differences between Neurostat 3D-SSP and syngo.via Database Comparison. These findings were not surprising as the latter software is based on Neurostat, adding a slice by slice display in the three projections. The detectability of SPM improved after using *p* < 0.005 rather than *p* < 0.001 as a threshold value. However, by decreasing the level of statistical significance, there is a risk of a subsequent increase of statistical noise, which may hinder the evaluation of images. The advantages of SSP methods are rapid post-processing and the availability of normal databases stratified by age groups. In contrast, SPM can be used in a 15O-water PET perfusion database.

In our case series, hypermetabolic or hypometabolic findings showed different patterns according to the specific antibodies involved. Namely, LGI-1 and CASPR2 (previously known as VGKC) had medial temporal involvement as it has been previously described [6]. Both anti-LGI-1 cases described herein showed cerebellar and basal ganglia hypermetabolism with frontal hypometabolism, in line with previous reports [12]. Similarly, the case with anti-NMDA receptor encephalitis and the two anti-CASPR2 cases revealed previously described patterns. This pattern specificity has not been found in two large cohort-studies [26,27] probably due to a longer time gap between the clinical onset and the FDG-PET imaging. In both studies, the FDG-PET was performed during the diagnostic period, although the time from the onset of symptoms to the FDG-PET was up to 4 weeks [27], as opposed to 1 week in our study. These cohort studies found no pattern differences across age groups, antibody type or AE classification.

We detected FDG-PET abnormal findings in patients with normal MRI (2/6). In fact, MRI was unremarkable for one month after the onset of symptoms in the seronegative case, showing higher sensitivity of FDG-PET. Additionally, when using initial MRI, CSF analysis and EEG, FDG-PET was positive in one case when all the remaining tests were normal.

As previously documented, brain FDG-PET scans may show pathological findings in cases with normal MRI. This is more significant for anti-NMDA receptor encephalitis [12], for which MRI is not included among the diagnostic criteria [3]. In line with the available literature [3,26,28,29], we found no brain MRI abnormalities.

Probasco and colleagues [27] performed a retrospective analysis of 32 patients with autoantibody positive AE using a voxel-based analysis, with a commercially available database of over 250 age-stratified healthy controls, CortexID (GE Healthcare) based on the 3D-SSP method [23]. In their study, FDG-PET was abnormal in 52/61 (82%) of cases, while MRI alterations were observed in 40% of cases, CSF inflammation was detected in 62% of cases and the initial EEG was abnormal in 30% of cases. Despite these remarkable results, the lower FDG-PET sensitivity may be explained by the different onset-to-scan time gap compared to our series.

Newey and colleagues [30], reported six patients who underwent an FDG-PET (five positive anti-VGKC, one positive anti-NMDA). CSF analysis was abnormal in three patients, the EEG was reported as normal in one patient and the MRI was negative in the patient with positive anti-NMDA receptor. The use of voxel-based analysis was not specified. These findings are in line with our series, where three cases showed MTL hypermetabolism prior to MRI alterations.

The differential diagnosis of AE includes many different disorders such as CNS infections, namely, herpes virus encephalitis, primary CNS angiitis, acute disseminated encephalomyelitis, Susac’s syndrome and prion disease as well as Hashimoto’s encephalopathy (HE) [3]. In the latter, the initial brain MRI is characteristically normal, whereas the other entities are usually associated with MRI abnormalities. HE diagnosis, however, is based on the presence of antithyroid antibodies, the most important being anti-TPO. Recent evidence suggests TPO antibodies are not specific and do not predict responsiveness to steroids, which is believed to be the gold standard treatment for this disorder. HE has neither a specific biomarker nor typical neuropathologic findings [31]. FDG-PET in suspected HE, a different type of AE, may be of use as suggested by a recent report [32].

Our study is limited by the small number of cases, which does not allow description of new patterns associated with autoantibodies. However, the patterns we found are in line with what has been described in literature as characteristic or more frequent for each autoantibody [11,12,13,14,15,16,17,18,19]. Another limitation is that EEG recording was not performed at the time of the FDG injection, so we cannot exclude that some of the findings may be related to the epileptic activity.

Most reports to date lack statistical power and, therefore, enough reliability. Not only larger validation studies are needed, but also more objective semi-quantitative measures. As outlined by the European Association of Nuclear Medicine [33], these specific semi-automated approaches to analyze FDG-PET data were developed for Alzheimer’s disease and therefore are not equally suitable for identifying hypermetabolic areas. Likewise, they pointed out that some voxel-based approaches, depending on the choice of the reference region for intensity normalization, may lead to biased hypermetabolic areas. Consequently, a better standardization of FDG-PET reading could help to establish the role for FDG-PET in the diagnostic workup of autoimmune encephalitis.

## 5. Conclusions

For the evaluation of patients with suspected AE, standard analysis of FDG-PET images benefits from voxel-based analysis, as it may lead to more comparable and accurate results. This study provides new evidence of the utility of FDG-PET for AE beyond the approach based on MRI, CSF sampling and EEG. Patients with AE may benefit from prompt diagnosis when brain FDG-PET is added to the traditional complementary tests. Multicenter studies with larger series are warranted to evaluate and generalize voxel-based analysis in defining specific patterns and helping the clinical diagnosis of AE.

## Figures and Tables

**Figure 1 diagnostics-10-00356-f001:**
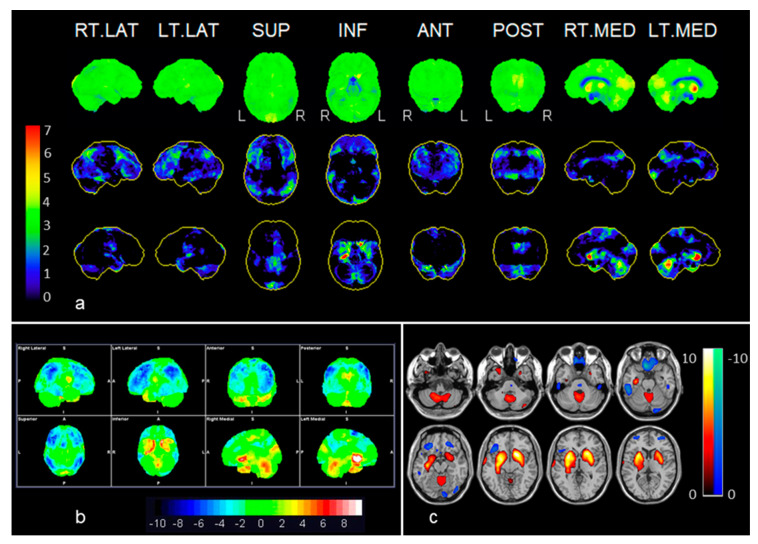
Example of anti-LGI-1 (case 2): (**a**) Neurostat: the first row shows surface projections of brain metabolism (visual assessment); the second row shows significant decreases in brain metabolism (red to green); and the third row shows significant increases (red to green) in brain metabolism as compared to an adjusted normal database. (**b**) syngo.via Database Comparison, and (**c**) Statistical Parametric Mapping (SPM 12). Statistical surface projections using Neurostat (**a**) and syngo.via Database Comparison (**b**) assessment distinguished better than SPM the frontal, lateral temporal and parietal hypometabolism, whereas hypermetabolic areas involving basal ganglia, cerebellar vermis and the medial aspect of the right temporal lobe were seen by the three methods. Color bars represent significant increases or decreases in brain metabolism compared to a normal database stratified by age. In the displayed Neurostat and SPM results, all the colored voxels represent statistical significance when compared to normal controls (Neurostat: increases and decreases in red to green; SPM: increases in, red to yellow, decreases in blue) whereas in the syngo.via Database Comparison, the significant voxels can be identified according to the SD color bar, 2 SD being the threshold of statistical significance (i.e., areas with green voxels are not significant). See also Table 2.

**Figure 2 diagnostics-10-00356-f002:**
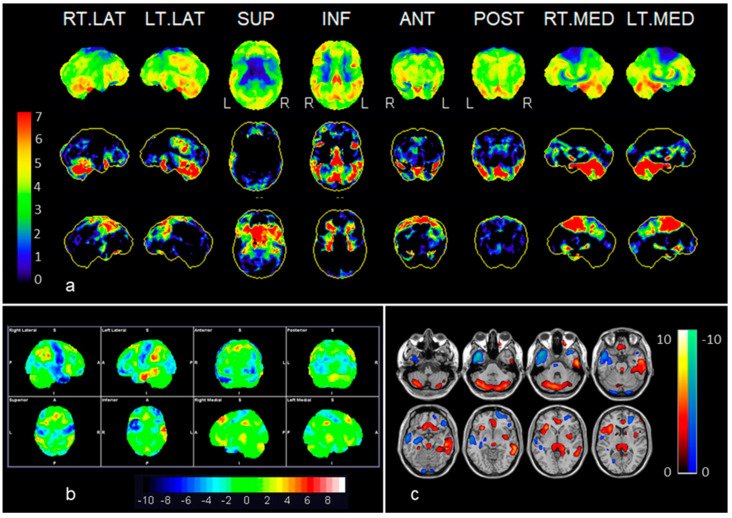
Example of anti-NMDAR (case 1): Neurostat (**a**) and syngo.via Database Comparison (**b**) were superior to SPM (**c**) exhibiting bilateral frontal, right temporal, occipital and bilateral motor cortex hypometabolism, as well as more hypermetabolic areas including temporo-parietal, posterior cingulate and cerebellum. Color bars represent significant increases or decreases in brain metabolism compared to a normal database stratified by age. In the displayed Neurostat and SPM results, all the colored voxels represent statistical significance when compared to normal controls (Neurostat: increases and decreases in red to green; SPM: increases in, red to yellow, decreases in blue), whereas in the syngo.via Database Comparison, the significant voxels can be identified according to the SD color bar, 2 SD being the threshold of statistical significance (i.e., areas with green voxels are not significant). See also Table 2.

**Figure 3 diagnostics-10-00356-f003:**
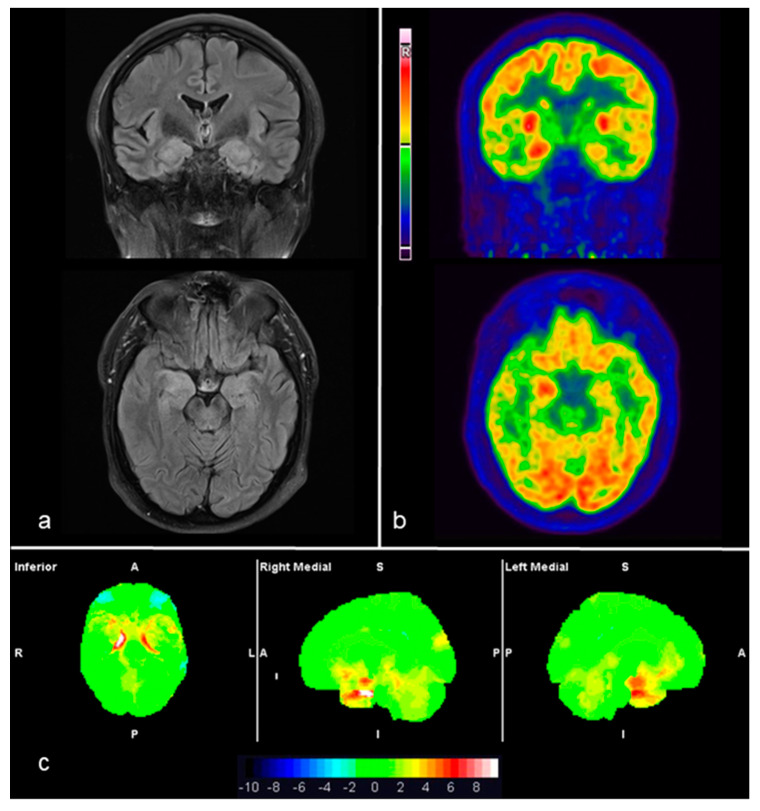
Example of anti-CASPR2 (case 3): MRI-3T, T2-FLAIR axial and coronal slices (**a**) show medial temporal hyperintensity as well as swelling, predominantly on the right side. Corresponding 18F-FDG-PET/CT axial and coronal slices (**b**) showing right medial temporal hyperactivity. The syngo.via Database Comparison based analysis of FDG-PET images (**c**) exhibited a significant bilateral medial temporal hypermetabolism on the statistical surface projection (in red) with respect to a normal database. In the displayed syngo.via Database Comparison, the significant voxels can be identified according to the SD color bar, 2 SD being the threshold of statistical significance (i.e., areas with green voxels are not significant and areas with orange to red voxels are significant increases). See also Table 2.

**Table 1 diagnostics-10-00356-t001:** Clinical data and tests.

Case	Age	Gender	Antibodies Type	Cognitive Impairment	Behavioral Disorder	Seizures	Treatment	Improvement after Treatment	EEG	CSF *	PET Result	MRI Result	Additional
1	17	F	NMDAR	Yes	Yes	Yes	Ig + MP	Yes	+	+	abnormal	normal	-
2	74	M	LGI-1	Yes	No	Yes	Ig + MP + rituximab	Yes	+	+	abnormal	abnormal	hyponatremia
3	65	M	CASPR2	Yes	No	Yes	Ig + MP + rituximab	Yes	+	+	abnormal	abnormal	-
4	77	F	LGI-1	Yes	Yes	Yes	Ig + MP	Yes	+	−	abnormal	abnormal	hyponatremia
5	70	F	No	Yes	No	No	Ig + MP	Yes	−	−	abnormal	normal ***	-
6	78	M	CASPR2	Yes	Yes	Yes **	Ig + MP	Yes	−	−	abnormal	abnormal	-

CSF: cerebrospinal fluid; EEG: electroencephalogram; F: female; Ig + MP: immunoglobulins 0.4 g/kg/day and methylprednisolone 1 g/day for 5 days; M: male. * Abnormal CSF was mainly lymphocytic pleocytosis with normal glucose and mild elevation of proteins (>50 mg/dL). + Abnormal findings as described for autoimmune encephalitis; − normal findings as described for autoimmune encephalitis. ** The patient had episodes suggestive of autonomic seizures that were not monitored with EEG. *** Initial MRI was normal but a second MRI performed one month later was pathological.

**Table 2 diagnostics-10-00356-t002:** Brain FDG-PET analysis.

Case	Antibodies Type	Visual Assessment		SPM				Syngo.via Database Comparison		Neurostat 3D-SSP	
				*p* < 0.001		*p* < 0.005					
		Hypo	Hyper	Hypo	Hyper	Hypo	Hyper	Hypo	Hyper	Hypo	Hyper
1	**NMDAR**	L Frontal, L&R Temporal, Occipital, Motor cortex	L lateral Temporal	R MTL and lateral Temporal, L Frontal. R Motor cortex	R. Insula; L lateral Temporal, L&R Parietal; PC	Similar locations but more extended	Similar locations but more extended	L&R Frontal, R temporal, occipital, L&R Motor cortex	L temporal, medial Frontal, Insula, PC, L&R Parietal, Cerebellum	L&R Frontal, R temporal, occipital, L&R Motor cortex	L Temporal, medial Frontal, Insula, PC, L&R Parietal, Cerebellum
2	**LGI-1**	Frontal, Parietal, Temporal, Thalamus, Occipital.	L&R BC	Orbitofrontal, L Temporal	L&R BG, Cerebellar vermis; L&R MTL	Similar locations and R Parietal	Similar locations but more extended	L&R Lateral Frontal, L Temporal, L&R Parietal, R PC	L&R BG, Cerebellar vermis, L&R MTL	L&R Lateral Frontal, lateral Temporal, Parietal, PC	L&R BG, Cerebellar vermis, L&R MTL
3	**CASPR2**	-	R. MTL R. BG	-	R MTL, R BG	L&R Frontal, R. Temporal	L&R MTL; R BG; Occipital	Frontal, R Temporo-Parietal	L&R MTL, R BG, Occipital	R. Frontal	L&R MTL
4	**LGI-1**	L Frontal, L&R parietal	L&R MTL, Cerebellar vermis, R BG	L&R Frontal, L&R lateral Temporal, R Parietal, L PC	L MTL	Similar but more extended, L&R Parietal, L&R PC	L&R MTL	L&R Frontal, L&R Parietal, L&R PC	L&R MTL Cerebellar Vermis, L&R BG, L&R Motor cortex	L&R Frontal, R Parietal, L&R PC	L&R MTL, Cerebellar vermis, Motor cortex, L&R
5	**Negative**	L Frontal, L lateral Temporal	PreCuneus, Occipital	L&R Frontal, L&R Temporal	-	L&R Frontal, R Insula, L&R Temporal	R Parietal	L&R Frontal, L&R Parietal, L Temporal	Parieto-Occipital, Precuneus,	L&R Frontal	Parieto-Occipital
6	**CASPR2**	-	L.MTL.	L&R Fronto-temporal	-	Similar locations but more extended, Parietal	-	L&R Fronto-temporal	L MTL.	L&R Fronto-temporal	L MTL, Parieto-Occipital

Hyper: Hypermetabolism; Hypo: Hypometabolism; L: Left; R: Right; PC: Posterior cingulate; BC: Basal ganglia; MTL: Medial temporal lobe.

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
