# Peer review of "18F-FDG-PET Imaging Patterns in Autoimmune Encephalitis: Impact of Image Analysis on the Results"

_diagnostics, 2020, doi:10.3390/diagnostics10060356_

Round 1

Reviewer 1 Report

Moreno-Ajona et al present a study of 6 patients with auto-immune encephalitis and examine the diagnostic value of FDG-PET scans comparing visual inspection and computerised analyses.

The study is obviously limited by its small sample size and the heterogeneity of the cases reported. Consequently, the assertion about specific pattern linked with antibodies should be largely mitigated.

Several points should be more detailed. I could not find (maybe I am missing it) the delay between the onset of symptoms and FDG-PET scans. Similarly, were some of the people in ICU, maybe sedated? This could also have an impact on the comparison with healthy controls. Finally, most patients had interictal abnormalities on EEG, had they EEG recording during FDG injection to rule out that seizures might have participated to the hypersignals found? If not, this should be acknowledged as a limitation.

Author Response

Reviewer 1:

GENERAL COMMENT:

English language and style are fine/minor spell check required 

 RESPONSE: Thank you for addressing this. Indeed, the manuscript was reviewed by a native English speaker and minor changes were made.

CHANGES IN THE MANUSCRIPT: Changes have been made to reflect English language/style and spelling changes. 

COMMENT 1: Moreno-Ajona et al present a study of 6 patients with auto-immune encephalitis and examine the diagnostic value of FDG-PET scans comparing visual inspection and computerised analyses.

The study is obviously limited by its small sample size and the heterogeneity of the cases reported. Consequently, the assertion about specific pattern linked with antibodies should be largely mitigated.

 RESPONSE: We thank the reviewer for this appraisal. The strength of this case series relies on the detailed description of different methods of analysis. We understand the concerns about the small number of subjects and heterogeneity of the cases included in the study. We have added this as a limitation at the end of the Discussion. We agree with the reviewer that it is not possible to consistently determine new patterns associated with autoantibodies because of the small number of cases.  Nevertheless, the patterns we found were in line with what had been described in the literature as characteristic or more frequent for each autoantibody.

CHANGES IN THE MANUSCRIPT: The Discussion, now reads “Our study is limited by the small number of cases which does not allow description of new patterns associated with autoantibodies. However, the patterns we found are in line with what has been described in the literature as characteristic or more frequent for each autoantibody [11-19]”.

 COMMENT 2: Several points should be more detailed. I could not find (maybe I am missing it) the delay between the onset of symptoms and FDG-PET scans.

 RESPONSE: Thank you for this point. This is included in the “Material and Methods” section, under the headline “2.1. Database analysis” where it reads 18F-FDG brain PET study was performed in all cases within the first week of symptoms’ onset”.

This is not commented further throughout the manuscript which we believe would be of use in the Discussion, where comparison to other studies is made on this respect.

CHANGES IN THE MANUSCRIPT: The Discussion, now reads “In both studies, the FDG-PET was performed during the diagnostic period, although the time from the onset of symptoms to FDG-PET was up to 4 weeks [27], as compared to up to 1 week in our study”.

 COMMENT 3. Similarly, were some of the people in ICU, maybe sedated? This could also have an impact on the comparison with healthy controls.

 RESPONSE: We are particularly grateful for this appraisal. At the moment of the FDG-PET scans, none of the cases were under sedatives or anesthetic agents. This could have hindered the results we obtained and we agree this should be mentioned in the manuscript for clarification. Similarly, treatment with antiepileptic drugs could potentially alter the results and, because the FDG-PET was completed as part of the initial workup, none of the cases presented were on these drugs when the scan was performed.

CHANGES IN THE MANUSCRIPT: The “Results” section, subsection “3.2. Brain 18F-FDG-PET/CT findings” now reads “Of note, none of the cases were on sedative, anesthetic or antiepileptic drugs at the time of the FDG-PET scan”.

COMMENT 4: Finally, most patients had interictal abnormalities on EEG, had they EEG recording during FDG injection to rule out that seizures might have participated to the hypersignals found? If not, this should be acknowledged as a limitation.

 RESPONSE: We thank the reviewer for this comment and we agree this is a limitation to our findings. As commented above, none of the cases were on antiepileptic drugs when the scans were performed and, as for the FDG-PET scan, the EEG was completed within the first week after the onset of symptoms. In fact, in some cases, the EEG was completed after the FDG-PET because of the good availability of tests in our site. A downside to the absence of treatment with antiepileptic drugs is the higher likelihood of the occurrence of a seizure during the FDG-PET scan. EEG recording was NOT performed at the moment of the FDG injection so we cannot exclude some of the findings are related to the epileptic activity. In line with the reviewer’s opinion, we agree hypermetabolism may be partly related not only with inflammation but also with the epileptic disease. However, none of the cases had clinical symptoms of seizures when the scans were performed, none had electrophysiological signs of non-convulsive status epilepticus and interictal findings were discharges on the temporal areas but not delta waves which would reasonably correlate with hypometabolic areas.

CHANGES IN THE MANUSCRIPT: The Discussion, now reads “Another limitation is that EEG recording was not performed at the time of the FDG injection so we cannot exclude some of the findings may be related to the epileptic activity”.

REFERENCES

  1. Minoshima S, Frey KA, Foster NL, Kuhl DE. Preserved pontine glucose metabolism in Alzheimer disease: a reference region for functional brain image (PET) analysis. J Comput Assist Tomogr 1995;19:541-547.
  2. Karow DS, McEvoy LK, Fennema-Notestine C, et al. Relative capability of MR imaging and FDG PET to depict changes associated with prodromal and early Alzheimer disease. Radiology 2010;256:932-942.
  3. Marcoux A, Burgos N, Bertrand A, et al. An Automated Pipeline for the Analysis of PET Data on the Cortical Surface. Front Neuroinform 2018;12:94.

Reviewer 2 Report

This manuscript presents a small, mixed cohort of 6 patients with autoimmune encephalitis (AE), for whom retrospective evaluation of the utility of brain 18F-FDG-PET in diagnosing AE, and evaluation of the additive value of voxel based analysis of 18F-FDG-PET with comparison to normal database as compared to standard visual evaluation, was performed.

The presented findings of the hypo- and hypermetabolic patterns in the acute phase of the various subtypes of AE are not novel per se.  However, the value of this report is in its attempt to evaluate the improved sensitivity of voxel based (semiautomated) methods for analysing brain 18F-FDG-PET when compared to standard visual analysis. On the other hand, the shortcoming in this respect is the heterogeneous and small sample size, although significant differences were reported already at individual level. Thus, pursuing to report such findings as a short report is justified.

However, some major concerns still remain:

Firstly, no validation or evaluation of the methodological robustness of performing the intensity normalization to pons has been provided (Page 3, line 113). It is strongly recommended to provide a valid reference for this, or to provide data evaluating the uptake of 18-F-FDG in pons in these AE patients vs. HC to rule out any bias in the comparisons to healthy database arising from this normalisation approach.

Further, the authors should provide better rationale for reporting p-values with both p<0.001 and p<0.005 in the SPM analyses (Table 2 and lines 167-170), or state that this was an exploratory approach.

In addition, the manuscript would benefit of overall language check for grammar errors and better readability.

Some minor points:

Figure 1 and 2: The figures looks slightly busy. Consider showing less views/slices, or enlarging the image panel.  Recommended to provide color scale bars for the images with respective explanations in Figure legends.

Figure 3: Please provide color scale bars for the images with respective explanations in Figure legends.

Discussion: It is recommended to discuss in general the differential diagnostic possibilites for suspected AE cases with normal initial MRI and abnormal FDG-PET to increase the clinical value of the report.

Page 9, rows 219-221: the sentence does not read very well, it is recommended to check the grammar. Also check the the nomenclature of syngo.PET  throughout the manuscript (several versions for the name of the software used in the manuscript: syngo.PET Neuro Database Comparison, Syngo Via Database and Syngo Database comparison)

Author Response

Please open "letter addressing comments" document as contains a graph.

REVIEWER 2

GENERAL COMMENTS: This manuscript presents a small, mixed cohort of 6 patients with autoimmune encephalitis (AE), for whom retrospective evaluation of the utility of brain 18F-FDG-PET in diagnosing AE, and evaluation of the additive value of voxel based analysis of 18F-FDG-PET with comparison to normal database as compared to standard visual evaluation, was performed.

The presented findings of the hypo- and hypermetabolic patterns in the acute phase of the various subtypes of AE are not novel per se.  However, the value of this report is in its attempt to evaluate the improved sensitivity of voxel based (semiautomated) methods for analysing brain 18F-FDG-PET when compared to standard visual analysis. On the other hand, the shortcoming in this respect is the heterogeneous and small sample size, although significant differences were reported already at individual level. Thus, pursuing to report such findings as a short report is justified. 

However, some major concerns still remain. Firstly, no validation or evaluation of the methodological robustness of performing the intensity normalization to pons has been provided (Page 3, line 113). It is strongly recommended to provide a valid reference for this, or to provide data evaluating the uptake of 18-F-FDG in pons in these AE patients vs. HC to rule out any bias in the comparisons to healthy database arising from this normalisation approach.

 RESPONSE: We appreciate this comment, as this is crucial issue that we might have not clarified enough. It is clear that any voxel-based analysis requires an intensity normalization procedure. However, in the literature, several approaches have been used (global man, cerebellum, pons…) and all of them have drawbacks. 

In this case-series, we preferred normalization to a particular region instead of global mean normalization due to the fact that encephalitis could potentially affect a large region within the brain and therefore could bias the global mean value. Pons is a region that has been widely used for this kind of analysis [1-3] and was chosen for this study because it was considered to be probably unaffected by the pathology under study. Following to the reviewer’s suggestion, we have verified that there were NO statistically significant differences in pons uptake between AE patients and HC (t-test, p=0.27).”.

CHANGES IN THE MANUSCRIPT: The “Material and Methods” section, subsection “2.2. FDG-PET image analysis” now reads “The pons is a region that has been widely used for activity normalization [23] and is considered to be unaffected by the pathology under study”.

 Further, the authors should provide better rationale for reporting p-values with both p<0.001 and p<0.005 in the SPM analyses (Table 2 and lines 167-170), or state that this was an exploratory approach.

RESPONSE: As pointed by the reviewer, these p-values were used as an exploratory approach to evaluate the regions with increases or decreases at two levels of significance.

CHANGES IN THE MANUSCRIPT: The “Material and Methods” section, subsection “2.2. FDG-PET image analysis”now reads “As an exploratory approach, the threshold of the T-map images was set at two significance levels, p <0.001 and p <0.005 (uncorrected) with an extent threshold of 40 voxels”.

 In addition, the manuscript would benefit of overall language check for grammar errors and better readability.

RESPONSE: Thank you for addressing this. Indeed, the manuscript was reviewed by a native English speaker and minor changes were made.

CHANGES IN THE MANUSCRIPT: Changes have been made to reflect English language/style and spelling changes. 

Figure 3: Please provide color scale bars for the images with respective explanations in Figure legends.

RESPONSE: We are grateful for this comment that will improve the understanding and interpretation of Figures.

CHANGES IN THE MANUSCRIPT: All the three figures and their corresponding figure legends have been modified by including the color scales of each method used (18F-FDG-PET visual analysis, Neurostat, syngo.via Database Comparison and SPM), and explanations.

 Discussion: It is recommended to discuss in general the differential diagnostic possibilites for suspected AE cases with normal initial MRI and abnormal FDG-PET to increase the clinical value of the report.

 RESPONSE: Thank you for this appraisal. The differential diagnosis of AE may include many different disorders such as CNS infections, namely herpes virus encephalitis, primary CNS angiitis, acute disseminated encephalomyelitis, Susac’s syndrome, prion disease as well as Hashimoto’s encephalopathy (HE) [4]. In the latter, the initial brain MRI is characteristically normal whereas the other entities are usually associated with MRI abnormalities. HE diagnosis, however, is based on the presence of antithyroid antibodies, the most important being anti-TPO. The problem is the existence of HE as such is still under debate. Recent evidence suggests TPO antibodies are not specific and do not predict responsiveness to steroids, which is believed to be the gold standard treatment for this disorder. HE has neither a specific biomarker nor typical neuropathologic findings [5]. We agree the differential diagnosis deserves further comments in the Discussion. Indeed, the use of FDG-PET in suspected HE may be of considerable interest [6].

 CHANGES IN THE MANUSCRIPT: The Discussion now reads “The differential diagnosis of AE includes many different disorders such as CNS infections, namely herpes virus encephalitis, primary CNS angiitis, acute disseminated encephalomyelitis, Susac’s syndrome, prion disease as well as Hashimoto’s encephalopathy (HE) (REF). In the latter, the initial brain MRI is characteristically normal whereas the other entities are usually associated with MRI abnormalities. HE diagnosis, however, is based on the presence of antithyroid antibodies, the most important being anti-TPO. Recent evidence suggests TPO antibodies are not specific and do not predict responsiveness to steroids, which is believed to be the gold standard treatment for this disorder. HE has neither a specific biomarker nor typical neuropathologic findings [REF]. FDG-PET in suspected HE, a different type of AE, may be of use as suggested by a recent report [REF]”.

Page 9, rows 219-221: the sentence does not read very well, it is recommended to check the grammar. 

 RESPONSE AND CHANGES IN THE MANUSCRIPT: We thank the reviewer for this comment. The sentence now reads “Voxel-based analyses detected MTL and extra limbic hypermetabolism, as well as hypometabolism, while the SSP methods were slightly more sensitive than SPM, but with no differences between Neurostat 3D-SSP and syngo.via Database Comparison”.

Also check the nomenclature of syngo.PET  throughout the manuscript (several versions for the name of the software used in the manuscript: syngo.PET Neuro Database Comparison, Syngo Via Database and Syngo Database comparison)
RESPONSE AND CHANGES IN THE MANUSCRIPT: Thank you for bringing this to our attention. The nomenclature is now “syngo.via Database Comparison”.

REFERENCES

  1. Minoshima S, Frey KA, Foster NL, Kuhl DE. Preserved pontine glucose metabolism in Alzheimer disease: a reference region for functional brain image (PET) analysis. J Comput Assist Tomogr 1995;19:541-547.
  2. 2. Karow DS, McEvoy LK, Fennema-Notestine C, et al. Relative capability of MR imaging and FDG PET to depict changes associated with prodromal and early Alzheimer disease. Radiology 2010;256:932-942.
  3. Marcoux A, Burgos N, Bertrand A, et al. An Automated Pipeline for the Analysis of PET Data on the Cortical Surface. Front Neuroinform 2018;12:94.
  4. Graus F, Titulaer MJ, Balu R, Benseler S, Bien CG, Cellucci T, et al. A clinical approach to diagnosis of autoimmune encephalitis. Lancet Neurol. 2016;15(4):391-404.
  5. Mattozzi S, Sabater L, Escudero D, et al. Hashimoto encephalopathy in the 21st century. Neurology 2020;94:e217-e224.
  6. Lagstrom RMB, Osterbye NN, Henriksen OM, Hogh P. Hashimoto's encephalopathy: Follow-up data from neuropsychology, lumbar puncture, and FDG-PET. Clin Case Rep 2019;7:1750-1753.

Round 2

Reviewer 1 Report

Changes made are appropriate

Author Response

Thank you for your comments and your help improving our manuscript. 

Reviewer 2 Report

The scientific content of this manuscript has improved in quality after revisions along the lines of the reviewer’s comments. In addition, the general readability has benefited from the English language check.

However, some minor issues and comments still remain:

Figures 1-3: It appears, that in the displayed Neurostat and SPM results, all the coloured voxels exhibit statistical significance when compared to normal controls, whereas in the syngo.via Database Comparison, the significant voxels can be identified only with the SD colour bar, presumably 2 SD being the level of significance (i.e. areas with green voxels are not significant). For easier interpretation of the visualiation, I would suggest to make this clearer for the reader in the figure legends.

In addition, some minor language and proof reading issues remain:

Row 25: Should this read ”assisted by voxel-based analyses”?

Row 30-31: Suggested to correct into: ”Two cases had anti-LGI1, one had anti-NMDA-R and two anti-CASPR2 antibodies, and one was seronegative”

Rows 74-75: Please rephrase the sentence: ”On the other hand, hypometabolism may be found when antibody-capping occurs and subsequent receptor internalization in the presence of cell surface antibodies”

Row 80-81: Please rephrase the sentence: ”The method of analyses utilized in the studies aforementioned were essentially two:”

Rows 120-121: I believe the word ”unaffected” is missing in place of the letter ”d” in this sentence: ”The pons is a region that has been widely used for activity normalization [23] and is considered to be d by the pathology under study.

Row 236: please remove the duplicate ”in the in the”

Row 348: ”guaranteed”; should this be ”warranted”?

Author Response

 Comments and Suggestions for Authors

The scientific content of this manuscript has improved in quality after revisions along the lines of the reviewer’s comments. In addition, the general readability has benefited from the English language check.

  • We thank the reviewer for their generous comment and, indeed, for a thorough review, taking the time to suggest the following edits.

However, some minor issues and comments still remain:

Figures 1-3: It appears, that in the displayed Neurostat and SPM results, all the coloured voxels exhibit statistical significance when compared to normal controls, whereas in the syngo.via Database Comparison, the significant voxels can be identified only with the SD colour bar, presumably 2 SD being the level of significance (i.e. areas with green voxels are not significant). For easier interpretation of the visualiation, I would suggest to make this clearer for the reader in the figure legends.

  • We thank the reviewer for this comment and we agree that by amending this we add clarity to the reading of the figures 1-3. We have added the following paragraph to figure legends 1-3: “Color bars represent significant increases or decreases in brain metabolism compared to a normal database stratified by age. In the displayed Neurostat and SPM results, all the colored voxels represent statistical significance when compared to normal controls, whereas in the syngo.via Database Comparison, the significant voxels can be identified according to the SD color bar, being 2 SD the threshold of statistical significance (i.e. areas with green voxels are not significant)”.

In addition, some minor language and proof reading issues remain:

Row 25: Should this read ”assisted by voxel-based analyses”?

Row 30-31: Suggested to correct into: ”Two cases had anti-LGI1, one had anti-NMDA-R and two anti-CASPR2 antibodies, and one was seronegative”

Rows 74-75: Please rephrase the sentence: ”On the other hand, hypometabolism may be found when antibody-capping occurs and subsequent receptor internalization in the presence of cell surface antibodies”

Row 80-81: Please rephrase the sentence: ”The method of analyses utilized in the studies aforementioned were essentially two:”

Rows 120-121: I believe the word ”unaffected” is missing in place of the letter ”d” in this sentence: ”The pons is a region that has been widely used for activity normalization [23] and is considered to be d by the pathology under study.

Row 236: please remove the duplicate ”in the in the”

Row 348: ”guaranteed”; should this be ”warranted”? 

  • Thank you so much for pointing out these stylistics, grammatical errors and typos that we have amended as suggested.